ecology/computational biology/
theoretical biology

biodiversity change, non-equilibrium ecology,
eco-evolutionary dynamics, stochastic differential
equation, species extinction

**Author for correspondence:**
Tsung-Jen Shen
e-mail: tjshen@nchu.edu.tw

†These authors contributed equally to this study.

# Extinction debt in local habitats: quantifying the roles of random drift, immigration and emigration

Yongbin Wu[1,†], Youhua Chen[2,†], Shui-Ching Chang[3], You-Fang Chen[2] and Tsung-Jen Shen[3]

[1]College of Forestry and Landscape Architecture, South China Agricultural University, Guangzhou, Guangdong, China
[2]CAS Key Laboratory of Mountain Ecological Restoration and Bioresource Utilization, and Ecological Restoration and Biodiversity Conservation Key Laboratory of Sichuan Province, Chengdu Institute of Biology, Chinese Academy of Sciences, Chengdu 610041, China
[3]Institute of Statistics and Department of Applied Mathematics, National Chung Hsing University, 250 Kuo Kuang Road, Taichung 40227, Taiwan

T-JS, 0000-0002-1742-9730

We developed a time-dependent stochastic neutral model for predicting diverse temporal trajectories of biodiversity change in response to ecological disturbance (i.e. habitat destruction) and dispersal dynamic (i.e. emigration and immigration). The model is general and predicts how transition behaviours of extinction may accumulate according to a different combination of random drift, immigration rate, emigration rate and the degree of habitat destruction. We show that immigration, emigration, the areal size of the destroyed habitat and initial species abundance distribution (SAD) can impact the total biodiversity loss in an intact local area. Among these, the SAD plays the most deterministic role, as it directly determines the initial species richness in the local target area. By contrast, immigration was found to slow down total biodiversity loss and can drive the emergence of species credits (i.e. a gain of species) over time. However, the emigration process would increase the extinction risk of species and accelerate biodiversity loss. Finally but notably, we found that a shift in the emigration rate after a habitat destruction event may be a new mechanism to generate species credits.

## 1. Introduction

Predicting biodiversity change because of climate forcing and habitat loss is a cornerstone in the research of contemporary community ecology and conservation biology. However, because

species loss is a time-delayed process that is usually described as extinction debts [1–4], many profound ecological problems are still poorly known and remain open for exploration, including: (i) How fast will species go extinct due to abrupt habitat destruction? (ii) To which factors will the velocity of species extinction be correlated? (iii) Can we accurately predict the temporal trajectory of species loss? (iv) How will community patterns (e.g. species abundance distributions) be altered if species loss is inevitable?

Many previous studies attempted to demystify extinction debts using a variety of approaches. For example, several previous studies [5,6] used 'static' (i.e. time-irrelevant) approaches to model delayed extinction patterns caused by habitat destruction. However, such methods are unable to characterize the temporal trajectories of species extinctions, which are clearly time-dependent. In comparison, inspired by the theory of island biogeography, some other papers developed dynamic models (mostly related to species–area relationships) to predict the delayed loss of species richness [7–10]. However, these methods typically require an estimation of a relaxation parameter, which is usually difficult to fit [7] and has few empirical values to which one can refer [11]. Moreover, other than species richness, community-level delayed consequences (e.g. compositional change: how abundant and rare species will change) cannot be clearly demonstrated using these dynamic models.

Neutral theory and models can be powerful for quantifying delayed consequences of habitat loss on biodiversity. By assuming the functional equivalence of different species subject to stochastic birth, death and dispersal processes, neutral models are able to quantify how individual species and the entire ecological community will respond to habitat destruction [12–14]. However, no previous studies have systematically employed and developed neutral models to address the above-mentioned questions.

In this study, we developed a general neutral model that can predict the transitional behaviours of species extinctions by incorporating multiple essential ecological mechanisms: emigration/immigration, random drift and habitat destruction. More importantly, compared to previous studies, our model can explicitly project how community patterns will shift from both spatial and temporal perspectives.

# 2. Material and methods

## 2.1. A stochastic dispersal–birth–death model

Here, we use the following stochastic dispersal–birth–death (SBD) model to describe the temporal dynamics of a species' population under random drift, immigration and emigration processes

$$\left.\begin{aligned}
\frac{\mathrm{d}p_0(t)}{\mathrm{d}t} &= p_1(t) - vp_0(t) \\
\frac{\mathrm{d}p_1(t)}{\mathrm{d}t} &= vp_0(t) + 2p_2(t) - (2-u)p_1(t) \\
&\vdots \\
\frac{\mathrm{d}p_j(t)}{\mathrm{d}t} &= (j+1)p_{j+1}(t) + (j-1)(1-u)p_{j-1}(t) - j(2-u)p_j(t).
\end{aligned}\right\}
\tag{2.1}$$

Note that $p_j(t)$ in equation (2.1) is the probability that a given species has abundance $j$ at time $t$. We treat $v$ as the immigration rate and $u$ as the emigration rate. The immigration rate is necessary for the model; otherwise, no species will exist in the study area, and community dynamics are not possible if the area originally had no organisms in existence, and no individuals could disperse into the area from the outside. A flowchart for showing the transition dynamic of neighbouring abundance states from equation (2.1) is visually illustrated in electronic supplementary material, figure S1A of the appendix. This model only allows the flow of immigration when there are no individuals inside the target area ($vp_0(t)$). This assumption seems a bit strong, but there are a variety of reasons for assuming so (details are introduced in the Discussion section). To further demonstrate the usefulness of the proposed model (equation (2.1) as to the evaluation of stochastic dispersal process in population dynamic and extinction, we also introduce and compare a full immigration model [15,16], detail of which is shown in the Additional Methods of the appendix (electronic supplementary material, figure S1B and equation S1). In that full immigration model, immigration can take place as long as the population of species changes in an increasing direction (electronic supplementary material, figure S1B).

By defining the probability generating function (PGF) as

$$H(z,t) = \sum_{n=0}^{\infty} z^n p_n(t),$$

equation (2.1) can be transferred to a first-order linear partial differential equation (PDE), as

$$\frac{\partial H(z,t)}{\partial t} = (1-z)(1-(1-u)z)\frac{\partial H(z,t)}{\partial z} + v(z-1)H(0,t). \tag{2.2a}$$

When there is no immigration effect (i.e. $v = 0$), the above solution can be recovered to a previously described one [17], and there is an analytical solution for the PGF $H(z,t)$, which has a form as follows [17]:

$$H(z,t) = \begin{cases} \left(\dfrac{1-(1-u)z-(1-z)e^{-ut}}{1-(1-u)(z+(1-z)e^{-ut})}\right)^{j}, & u > 0; \\[3mm] \left(\dfrac{t-tz+z}{t-tz+1}\right)^{j}, & u = 0, \end{cases} \tag{2.2b}$$

for the specific initial condition $p_n(0) = \delta(n-j)$ ($j \geq 0$), where $\delta(w) = 1$ if $w = 0$; and $\delta(w) = 0$ if $w \neq 0$. The probability of observing $n$ individuals at time $t$ can be derived by taking the $n$-ordered derivatives of the PGF as

$$p_n(t) = \frac{1}{n!}\frac{\partial^n H(z,t)}{\partial z^n}\bigg|_{z=0}. $$

However, it becomes challenging to analytically solve the PDE (equation (2.2a)) using the traditional method of characteristics when the immigration rate is not zero (i.e. $v \neq 0$), since the rightmost term of equation (2.2a) still involves $H(0,t)$. To this end, we use a matrix exponential method [18,19] when conducting numerical analyses. Specifically, we use the following matrix to record the coefficients of the ordinary differential equation (ODE) system in equation (2.1) as

$$G_m = \begin{pmatrix} -v & 1 \\ v & -(2-u) & 2 \\ & (1-u) & -2(2-u) & 3 \\ & & 2(1-u) & -3(2-u) & 4 \\ & & & 3(1-u) & -4(2-u) & 5 \\ & & & & \cdots & \cdots & \cdots \\ & & & & & (m-2)(1-u) & -(m-1)(2-u) & m \\ & & & & & & (m-1)(1-u) & -m(2-u) \end{pmatrix}, \tag{2.3}$$

and a column vector to record the time-dependent probability at each abundance state as

$$X_m(t) = (p_0(t), p_1(t), p_2(t), \ldots, p_m(t)). $$

For numerical computing purpose, we convert the infinite ODE system equation (2.1) into a finite ODE system with $m+1$ states (i.e. from 0 to a maximal abundance value denoted by $m$; integer $m > 1$). Therefore, equation (2.1) can be re-formulated as the following finite matrix form

$$\frac{dX_m(t)}{dt} = G_m X_m(0). \tag{2.4}$$

Accordingly, we can numerically solve the time-dependent probabilities by using matrix exponential method as

$$X_m(t) = \exp(G_m t) X_m(0). \tag{2.5}$$

With the result from equation (2.5), the time-dependent probability of extinction can be easily extracted from $X_m(t)$, i.e. its first element ($p_0(t)$). As a remark, because the original stochastic system in equation (2.1) is infinite, it is necessary to check the convergence of the derived extinction probability $p_0(t)$ for different matrix sizes ($m$) under the finite setting.

## 2.2. Total biodiversity loss at the metacommunity level

The probability that a species can still survive at time $t$ given that its initial abundance, $j$, under the above stochastic SBD process (equation (2.1)) is given by

$$p_{n>0}(t) = 1 - p_0(t). \tag{2.6}$$

Therefore, without both emigration and immigration effects ($u = v = 0$), equation (2.6) characterizes the effect of random shift on extinction and survival dynamic [12], which can be derived using equation (2.6) as $p_{n>0}(t) = 1 - (t/(1 + t))^j$. If the emigration rate of a species is not zero, but immigration is absent (i.e. $u > 0$ and $v = 0$), equation (2.6) returns to $p_{n>0}(t) = 1 - ((1 - e^{-ut})/(1 - (1 - u)e^{-ut}))^j$. Finally, if both emigration and immigration rates are not zero, we employ the matrix exponential method (equation (2.5)) to numerically evaluate its temporal patterns.

At the metacommunity level, suppose that there are $S(0 \mid A)$ species in the region with size $A$ at the initial time $t = 0$. Moreover, let $S_j(0 \mid A)$ denote the number of species with abundance $j$ at the initial time; then with no emigration or immigration effects ($u = 0$ and $v = 0$; under the pure random drift scenario), the expected species number at any time $t$ is given by

$$S(t \mid A) = \sum_{j=1}^{\infty} S_j(0 \mid A)(1 - p_0(t)). \tag{2.7}$$

Therefore, the expected number of individuals at time $t$ is

$$N(t \mid A) = \sum_{j=1}^{\infty} jS_j(t \mid A),$$

where $S_j(t \mid A) = S_j(0 \mid A)(1 - p_0(t))$ and represents the number of species with abundance $j$ in the region with size $A$ at time $t$. The expected total biodiversity loss at time $t$ at the metacommunity level, under either random drift or emigration processes, can be estimated as

$$E(t \mid A) = S(0 \mid A) - S(t \mid A). \tag{2.8}$$

When the immigration rate does not exist (i.e. $v = 0$), we can express $E(t \mid A)$ by

$$E(t \mid A) = \begin{cases} \sum_{j=1}^{\infty} S_j(0 \mid A)\left(\dfrac{t}{1 + t}\right)^j, & u = 0; \\ \sum_{j=1}^{\infty} S_j(0 \mid A)\left(\dfrac{1 - e^{-ut}}{1 - (1 - u)e^{-ut}}\right)^j, & u > 0. \end{cases} \tag{2.9}$$

This time-dependent solution for the total species loss at the metacommunity level is a concave function with respect to time $t$ when $u = 0$; however, the curvilinear pattern of $E(t \mid A)$ for the case $u > 0$ is indeterministic; one can refer to Additional Methods of the appendix in electronic supplementary material for a proof. By contrast, the species richness dynamic model presented in equation (2.4) of Sgardeli *et al.*'s [20] paper is a convex function in most cases when studying extinction debt [1,7,10,21]. A proof is also presented in electronic supplementary material, appendix. Accordingly, the derived biodiversity loss model in Sgardeli *et al.*'s [20] paper would become a concave function again when applying equation (2.8). Moreover, by properly assigning model parameters, both equation (2.8) and the model in Sgardeli *et al.*'s [20] paper will have similar prediction of temporal trajectory of species loss pattern (electronic supplementary material, figure S4). However, we have to mention here, when the immigration rate is not zero, the expected total biodiversity loss in our model (equation (2.8)) has to be solved out numerically.

## 2.3. Total biodiversity loss at the local community level

Suppose in the initial stage, there is a local area of size $a$ that is part of the region ($a \subset A$), and the local species abundance distribution (SAD) is given by $S_j(0 \mid a)$, which represents the number of species with abundance $j$ at the initial time in local area $a$. Accordingly, the expected total biodiversity loss at time $t$ at the local level, under random drift, immigration and emigration processes, can be estimated as

$$E(t \mid a) = S(0 \mid a) - S(t \mid a), \tag{2.10}$$

where $S(0 \mid a) = \sum_{j=1}^{\infty} S_j(0 \mid a)$; $S(t \mid a)$ represents the expected species number of the local area at any time $t$ and is estimated as

$$S(t \mid a) = \sum_{j=1}^{\infty} S_j(0 \mid a)p_{n>0}(t). \tag{2.11}$$

Note that this time-dependent solution of species loss at a local area (equation (2.10)) has a similar curvilinear pattern as remarked as to $E(t \mid A)$ in equation (2.9); one can refer to Additional Methods of the appendix in electronic supplementary material, for a proof.

## 2.4. An area-based Fisher's logseries model

To better incorporate instant habitat destruction on the transition behaviour of the species richness with Fisher's explicit statistical background, and accordingly, estimate the expected species loss over time, we assume there are $S(0\,|\,A) = 1000$ species at the initial time point $t = 0$ at the regional scale; and the SAD at either the regional or local scale follows an area-based logseries abundance model [22]. To be specific, the area-based model is given as follows.

Suppose a truncated negative binomial model is best fitted to the abundance distribution of all species necessarily present in target region $A$, and its probability mass function (PMF) is

$$P(N_A = n\,|\,A,k,\omega) = C\frac{\Gamma(k + n)}{\Gamma(k)\Gamma(n + 1)}\left(\frac{\omega}{\omega + A}\right)^k\left(\frac{A}{\omega + A}\right)^n, \quad n = 1,2\ldots, \tag{2.12}$$

where $C = (1 - (\omega/(\omega + A))^k)^{-1}$; $N_A$ is a random variate to depict the abundance of each species in the entire region $A$; $k$ and $\omega$ are two positive parameters of the PMF. Using this model with $k \to 0$, the limiting distribution can be derived by

$$\phi(n\,|\,A,\omega) = \lim_{k \to 0} P(N_A = n\,|\,A,k,\omega) = \left[\ln\left(1 + \frac{A}{\omega}\right)\right]^{-1}\left(\frac{1}{n}\right)\left(\frac{A}{\omega + A}\right)^n, \quad n = 1,2,\ldots, \tag{2.13}$$

where $x_A = A/(\omega + A)$ and $\alpha_A = [\ln(1 + A/\omega)]^{-1}$. This model can be further simplified to contain only one unknown parameter, $\alpha_A$, resulting in the following form:

$$\phi(n\,|\,\alpha_A) = \frac{\alpha_A}{n}(1 - e^{-1/\alpha_A})^n. \tag{2.14}$$

Suppose we only sample a local area of size $a$ from the entire region $A$. To do this, we define the number of individuals of each species observed in the local sampled area $a$ as $N_a$; then the PMF of $N_a$ is given by

$$\phi(n\,|\,\alpha_a,\alpha_A) = \lim_{k \to 0} P(N_a = n\,|\,a,A,k,\omega) = \frac{\alpha_A}{n}(1 - e^{-1/\alpha_a})^n, \quad n = 1,2,\ldots. \tag{2.15}$$

Detailed derivation of equation (2.15) can be found in our previous paper [22] and is thus omitted here. Therefore, the expected species richness with abundance $j$ in the local ecological community $a$ in the entire region $A$, given the parameter value $\alpha_A$, is given by

$$S_j(0\,|\,a,\alpha_a) = S(0\,|\,A) \times \phi(j\,|\,\alpha_a,\alpha_A).$$

As a result, the time-dependent and area-dependent total biodiversity loss of local sample $a$, before habitat destruction point $\tau$, can be computed as

$$E_a(t\,|\,t < \tau) = S_a(0) - S_a(t) = \sum_{j \geq 1} S_j(0\,|\,a,\alpha_a)p_0(t). \tag{2.16}$$

Suppose at some time point, $\tau$, a local area of size $b$ of region $A$ is instantly destroyed; then the number of extinct endemic species found in local area $b$ can be estimated by

$$E_b = S(0\,|\,A) \times \phi(0\,|\,\alpha_{A-b},\alpha_A).$$

Accordingly, the number of species found in the remaining habitat $(A - b)$ is given by

$$S_b = S(0\,|\,A) - E_b = S(0\,|\,A) \times (1 - \phi(0\,|\,\alpha_{A-b},\alpha_A)).$$

Then, the expected species richness with abundance $j$ in an intact local area, given the instantaneous destruction of another local area $b$ at time $\tau$, is given by

$$S_j(\tau\,|\,a,\alpha_a) = S_b \times \phi(j\,|\,\alpha_a,\alpha_A) = S(0\,|\,A)(1 - \phi(0\,|\,\alpha_{A-b},\alpha_A))\phi(j\,|\,\alpha_a,\alpha_A).$$

Therefore, the total biodiversity loss at local sample $a$, after the habitat destruction point $\tau$, becomes

$$E_a(t\,|\,t \geq \tau) = S_a(\tau) - S_a(t) = \sum_{j \geq 1} S_j(\tau\,|\,a,\alpha_a)p_0(t). \tag{2.17}$$

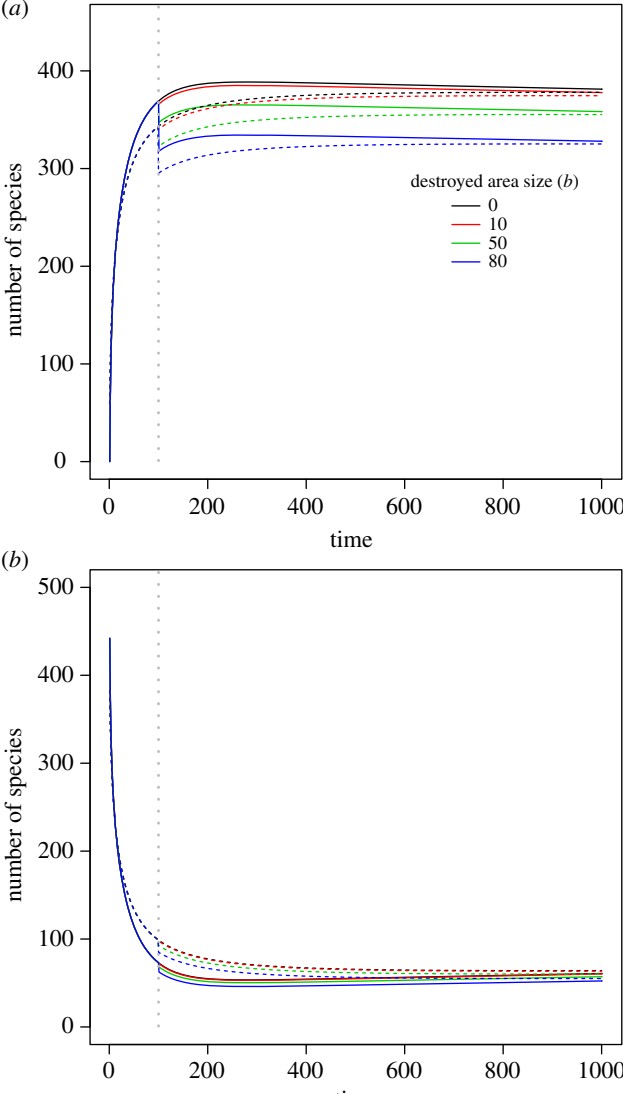

**Figure 1.** Relationship between biodiversity level and the size of the destroyed habitat. The local intact area of interest has a size of $a = 1$, while the total region has an area with a size of $A = 100$ and the initial species richness of $S(A|0) = 1000$. In subplot (*a*), the grey vertical dashed line indicates the time point at which the habitat destruction event takes place. Imminent extinction occurs at that time point, and accordingly, the biodiversity loss shows a sharp change at that point. Parameter $\omega$ in Fisher's logseries model is set to 0.001, the immigration rate $v$ is 0.02 while the emigration parameter $u$ is 0.01. Dashed curves are computed using a full immigration model (equation S1 of the appendix in electronic supplementary material). The *y*-axis denotes the difference of the number of species at the initial time and that at a specific time point afterwards. In subplot (*b*), *y*-axis denotes the number of species expected of the local intact area at a specific time point.

## 3. Results

By solving equation (2.4) numerically using equation (2.5), we can quantitatively evaluate the statistical behaviours of the infinite stochastic model presented in equation (2.1). To be specific, we found that starting with a higher state for initial abundance can slow down the rate of population extinction (electronic supplementary material, figure S2 of the appendix). Moreover, the immigration rate would retard the risk of species extinction, while the emigration rate would accelerate the extinction risk, as demonstrated from the diverse curve shape patterns of different combinations of immigration and emigration rates (electronic supplementary material, figure S3 of the appendix).

In the absence of habitat destruction, the total biodiversity loss dynamic tends to present a concave trend over time (figure 1). When an immediate habitat destruction event takes place, because of imminent extinction (i.e. loss of endemic species that are only found in the destroyed habitat), the curve is discontinuous (i.e. has an abrupt change) at the time point of habitat destruction (figure 1).

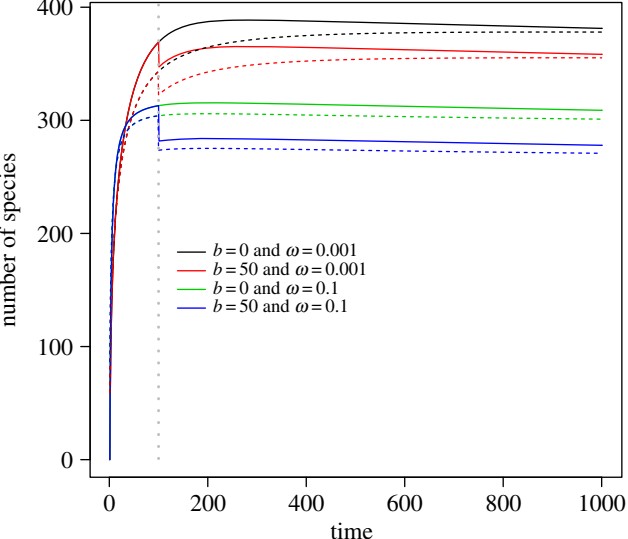

**Figure 2.** Joint impacts of the original species abundance distribution (SAD) and the degree of habitat destruction on the total biodiversity loss dynamic. Parameter $\omega$ controls the SAD curve shape in area-based logseries model. The grey vertical dashed line indicates the time point when the habitat destruction event takes place. The entire region has a total areal size of $A = 100$. The size of the area of the destroyed habitat was set to $b = 50$, while the local intact area of interest was set to a size of $a = 10$. The immigration rate is $v = 0.02$ while the emigration parameter is $u = 0.01$. Dashed curves are computed using a full immigration model (electronic supplementary material, equation S1 of the appendix). The y-axis denotes the difference of the number of species at the initial time and that at a specific time point afterwards.

After that, biodiversity loss continually accumulates and becomes smooth again, but the debt magnitude is always smaller than the original dynamic without habitat destruction (figure 1). The curve shape patterns were actually very similar between the proposed model (equation (2.1)) and the full immigration model (electronic supplementary material, equation S1).

High imminent extinction is expected when the areal size of the destroyed local habitat is great (figure 1). However, when the areal size of the destroyed habitat is higher, the expected total biodiversity loss after habitat destruction is lower (figure 1). This is because imminent extinction is paid off at the point in time when habitat destruction occurs.

The shape of the curve of the original SAD influences the shape of the curve of the total biodiversity loss dynamic (figure 2). When parameter $\omega$ in Fisher's logseries model is lower, the expected number of species in the local target area will be larger (electronic supplementary material, figure S5). In such a situation, total biodiversity loss over time will tend to be higher, and it takes a longer time to reach asymptotic values (figure 2). By contrast, when parameter $\omega$ in Fisher's logseries model is higher, the corresponding total biodiversity loss quickly becomes asymptotically stable because of low species richness in the target area (figure 2 and electronic supplementary material, figure S5). Note that the curvilinear trends were very similar between the proposed model (equation (2.1)) and the full immigration model (electronic supplementary material, equation S1).

When the emigration rate of a species is high, the expected total biodiversity loss is also high (figure 3). Moreover, for large emigration rates, 0.1 and 0.2, the time to equilibrium is very short as their expected total biodiversity losses keep unchanged after habitat destruction. However, the expected total biodiversity loss can increase first then decrease for the small emigration rate cases ($u = 0$ and 0.01). By contrast, increasing the immigration rate ($v$) can remarkably retard biodiversity loss in both the proposed (equation (2.1)) and the full stochastic immigration (electronic supplementary material, equation S1) models (figure 4). More interestingly, when looking at the temporal trajectory of species loss curve in figure 4, potential immigration credits can be observed (blue and green solid curves when $v = 0.1$ and 0.2, respectively). Analogous immigration credits were also observed for the full immigration model (blue and green dashed curves when $v = 0.1$ and 0.2, respectively, in figure 4).

Finally, when the intact area of interest has a larger areal size, the expected total biodiversity loss tends to be higher after habitat destruction (figure 5). However, this positive relationship is nonlinear. When the local intact area has a sufficiently large areal size, the total biodiversity loss tends to reach the maximal asymptotic value and only slightly increases when the size of the local target area is

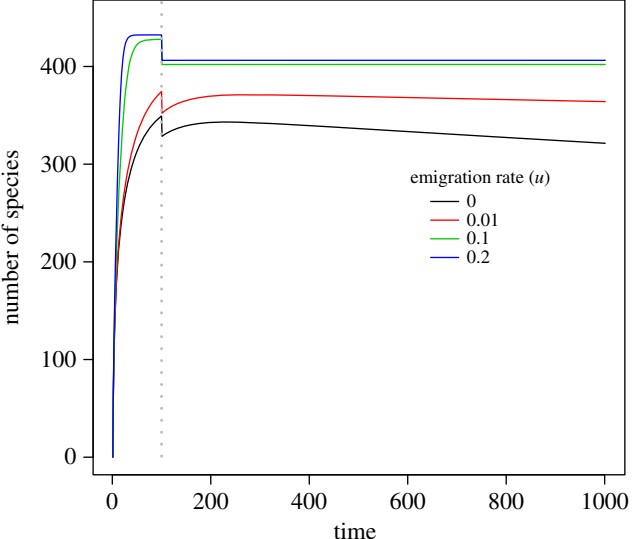

**Figure 3.** Impacts of the emigration rate ($u$) on the total biodiversity loss dynamic. The grey vertical dashed line indicates the time point when the habitat destruction event takes place. The entire region has a total areal size of $A = 100$. The areal size of the destroyed habitat was set to $b = 50$, while the local intact area of interest was set to a size of $a = 10$. Parameter $\omega$ in Fisher's logseries model was set to 0.001. In all the simulations, immigration rate ($v$) is set to be 0.01. The full immigration model is not compared here, because it does not take into account the emigration effect. The $y$-axis denotes the difference of the number of species at the initial time and that at a specific time point afterwards.

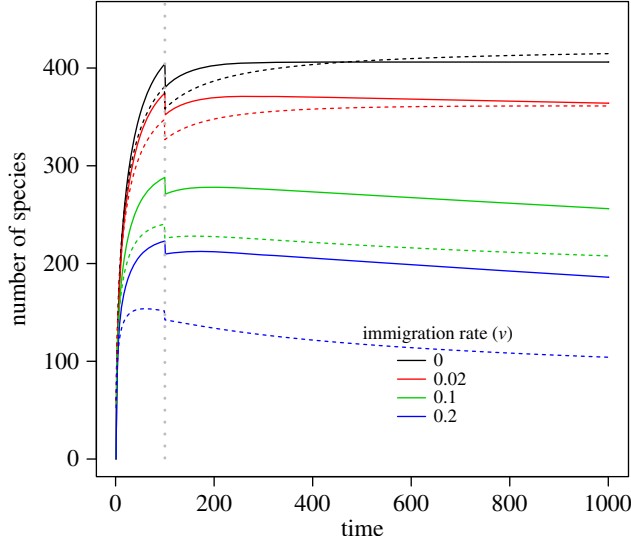

**Figure 4.** Impacts of the immigration rate ($v$) on the total biodiversity loss dynamic. The grey vertical dashed line indicates the time point when the habitat destruction event takes place. The entire region has a total areal size of 100. The areal size of the destroyed habitat was set to 50, while the local intact area of interest was set to a size of 10. Parameter $\omega$ in Fisher's logseries model was set to 0.001. In all the simulations, emigration rate is set to be 0.02. Dashed curves are computed using the full immigration model (electronic supplementary material, equation S1 of the appendix). The $y$-axis denotes the difference of the number of species at the initial time and that at a specific time point afterwards.

further increased. For the full immigration model, its curve shape patterns are very similar to those of our proposed model when comparing the solid and dashed lines in figure 5.

## 4. Discussion

Modelling biodiversity change, because it involves the coupling of spatial and temporal ecological factors, is an intriguing but challenging topic for global ecologists. Recent studies suggested that both

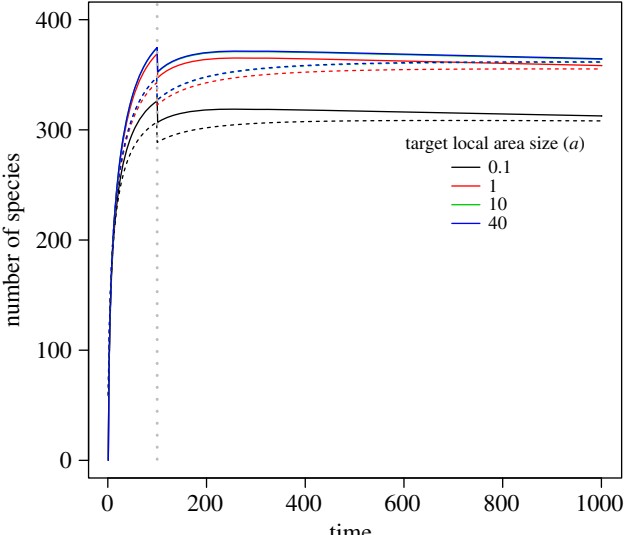

**Figure 5.** Relationship between the areal size of the local intact area of interest and the total biodiversity loss. The grey vertical dashed line indicates the time point when the habitat destruction event takes place. The entire region has a total areal size of $A = 100$. The areal size of the destroyed habitat was set to $b = 50$, while the local intact area of interest had varying sizes for comparison. Parameter $\omega$ in Fisher's logseries model was set to 0.001, the immigration rate $v$ is set to 0.02, and the emigration parameter $u$ is set to 0.01. Dashed curves are computed using the full immigration model (electronic supplementary material, equation S1 of the appendix). The y-axis denotes the difference of the number of species at the initial time and that at a specific time point afterwards.

ecological and evolutionary processes can take place on the same time scale [23,24]. To this end, our present model serves as a bridge to disentangle how ecological and evolutionary processes jointly explain the temporal complexity of the extinction of species.

In our study, we quantified temporal biodiversity loss pattern, which differs from previous studies that examined extinction debts. However, both concepts are closely related and become complementary to each other when discussing total extinction. That is, the sum of total biodiversity loss and extinction debt is equal to the total expected extinction when species richness reaches equilibrium after a sufficient time. As a consequence, our study actually already quantified temporal behaviours of species loss or the probability of extinction over time (electronic supplementary material, figure S3).

The parameter characterizing the immigration effect ($v$) can mitigate the potential biodiversity loss, while, by contrast, the parameter characterizing the emigration effect ($m$) would accelerate biodiversity loss (figures 3 versus 4). To this end, bidirectional dispersal processes (emigration versus immigration) could have great but contradictory impacts on ecosystem stability by accelerating versus reducing species extinctions, respectively (figures 3 and 4 and electronic supplementary material, figure S3). From a perspective of conservation biology, it is expected that the immigration effect represents a form of a rescue effect to maintain biodiversity in the studied local area by providing individuals of extinct species from the outside to the local community of interest. In our model (equation (2.1)), we assume that the arrival of a new immigrant is only possible when there are no individuals of the same species in the target community. This means, we assume that strict competitive exclusion effect exists between native individuals and immigrants, and the immigration event is possible when there are no native conspecifics in the community. However, native conspecifics (i.e. those individuals that are created by birth events) can coexist without exclusion.

This assumption seems a bit strong but realistic and a bit similar to the rescue effect in ecological literature [25–30]. In our paper, the rescue effect takes place through immigration from other areas only when the population size of a target species in a target area is sufficiently small. That is, the rescue effect ($v$) is magnified at most when there are no individuals of the target species in the target area. By contrast, the rescue effect ($v$) is ignored in our model (equation (2.1)) when there are some individuals of the target species in the focused area. This is because the dynamic of population is dominated by density-dependent growth and death rates (both rates = 1, see table 1 for details). The density-independent immigration rescue effect is therefore negligible, as $v$ is usually small (table 1).

**Table 1.** Assumptions used in the stochastic birth–death–immigration–emigration model of the present study (equation (2.1)).

| assumption | parameter value used or logistic answer |
| --- | --- |
| single species dynamic | yes |
| single location | yes |
| intrinsic growth rate | growth rate = 1 (this rate value is also used in Rosindell *et al.*'s [17] model) |
| intrinsic death rate | death rate = 1 (this rate value is also used in Rosindell *et al.*'s model) |
| density dependence | yes, density dependence is reflected by the fact that the changing rate of probability at abundance $j$ is dependent on the probability at abundance $j-1$, $j+1$ and $j$; and the associated magnitude $(j+1)$, $(j-1)(1-u)$ and $j(2-u)$. Such a kind of modelling density dependence in neutral models is also seen in previous studies [31,32] |
| interaction with other species | no, the model assumes the dynamic of a single species irrespective of the dynamic of other species |
| immigration rate | $v$, which is a small value in comparison to the intrinsic growth or death rate (which is equal to 1) in our model |
| emigration rate | $u$, which is a small value in comparison to the intrinsic growth or death rate (which is equal to 1) in our model |

Previous studies [27–30] working on the rescue effect also emphasized the importance of rescue effect for species that are extinction-prone and in small population sizes, akin to our model structure and assumption. However, the difference between theirs and ours is that we assume the rescue effect or immigration only takes place when there are no individuals of the species in the target area. Additionally, the model structure and assumption implemented in our paper are also a bit similar to the one-migrant-per-generation rule in conservation biology and genetics [33–35]. In our study, one migrant ($vp_0$) is only allowed for initiating the dynamic of a target population of a species in the studied area.

Finally but most importantly, even a more realistic immigration model is used (electronic supplementary material, equation S1 and figure S1B of the appendix), one can see that its resultant curve shape patterns were similar to those of our proposed model (figures 1, 2 and 4). Therefore, the proposed model (equation (2.1)) is sufficient for us to explore the potential roles of immigration versus emigration on biodiversity change over time. As summarized below, the present model reports the temporal trajectory of species loss, showing the emergence of immigration credits after habitat loss (figure 4) for the first time.

Previous studies extensively showed how spatial distributional patterns (particularly aggregation) can affect species extinctions at both the local and regional scales; when the spatial distributions of species vary from random to aggregated, either net losses or immigration credits are possible [6]. However, predictions of local or regional species losses or gains in those studies were debatable given that a clear statistical sampling framework in those papers was lacking. To resolve this statistical issue, the present study goes a step further to provide an area-based Fisher's model in which the local-regional sampling process is properly developed. Our numerical results showed that time-dependent immigration credits could occur when there is a mass of migrants from neighbouring regions (i.e. outside the targeted area; figure 4).

Furthermore, if species abundance distributions can be altered after habitat destruction, species gain might possibly be observed, as found in a previous study [6]. For the time-dependent model studied here, if the emigration rate of species is allowed to drop after habitat destruction, species gain is also possible (electronic supplementary material, figure S6). This phenomenon has never been observed in previous studies. To this end, we argue that a new process 'emigration credit', which differs from previous studies reiterating 'immigration credit' [3,4,6], is useful in explaining species gains due to historical habitat loss. That is, two independent dispersal processes, immigration and emigration, can contribute to the process of species gain. Our model is the first one to show, because of abrupt habitat destruction, how emigration can influence species loss and gain, and how immigration credit can emerge in local habitats over time, further extending previous static findings on the emergence of immigration credits in local habitats [6].

As discussed above, our conceptual framework is flexible, and the trajectory of biodiversity changes over time is directly relevant to the SAD (figure 2). This means that in addition to Fisher's logseries model studied in the present work, other ecologically interesting models of SAD (e.g. the Poisson lognormal model) can be incorporated and compared to evaluate how biodiversity will change over time. Furthermore, spatial aggregation patterns of species distributions might be incorporated as in previous studies [5,6,36–40]. Because our central goals were to integrate both ecological and evolutionary processes in modelling species extinction, we did not explicitly investigate how spatial distributional aggregation would influence temporal biodiversity changes. However, spatial aggregation is indeed an essential factor affecting the magnitude of extinction debt, because it is closely related to habitat fragmentation. Through numerical simulations, previous studies [37–40] have assessed the impacts of spatial percolation and scales on area-based estimation of species diversity and extinction. Given that traditional species–area relationships usually fail to capture the difference on the short- versus long-term species loss and the degree of habitat fragmentation [40,41], for future research, it becomes interesting to assess how complex spatial structures influence the temporal dynamic of species extinction.

Data accessibility. R code for generating figures 1 and 2 of main text is available from the Dryad Digital Repository: https://doi.org/10.5061/dryad.n710qb3 [42].
Authors' contributions. Y.W., Y.C. and T.-J.S. conceived of the study and designed the study, Y.W. and T.-J.S. coordinated the study, Y.C. and S.-C.C. helped derive the model, Y.-F.C. helped conduct the analyses, all the authors wrote and approved the final draft for submission.
Competing interests. We declare we have no competing interests.
Funding. This work was supported by the National Natural Science Foundation of China (grant no. 31901221), Strategic Priority Research Program of the Chinese Academy of Sciences (grant no. XDB31000000) and the Hundred Talents Program of the Chinese Academy of Sciences, Forestry Reform and Development Fund of the Central Government of China (grant no. [2019] GDTK-06) and Taiwan Ministry of Science and Technology (grant no. MOST 108-2118-M-005-002-MY2).
Acknowledgements. We thank the editors and two anonymous reviewers for spending their precious time to provide constructive comments that considerably help improve the quality of the study. We also thank Dr Qingdi Wang from the University of British Columbia for providing helpful advice on the numerical modelling of stochastic models. Partial work of this paper has been finished when Y.C. visited T.-J.S. in 2019; Y.C. appreciates the Mathematics Research Promotion Center (NSCMRPC; no. 108-25) to support his accommodations in part during the visit to Taiwan.

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
