## [Reviewer comments · Royal Society Open Science]

Review History

RSOS-191039.R0 (Original submission)

Review form: Reviewer 1

Is the manuscript scientifically sound in its present form?

No

Are the interpretations and conclusions justified by the results?

No

Is the language acceptable?

Yes

Do you have any ethical concerns with this paper?

No

Have you any concerns about statistical analyses in this paper?

No

Recommendation?

Reject

Comments to the Author(s)

In this submission, Chen et al. present a stochastic neutral model to estimate the biodiversity change under ecological disturbance and dispersal dynamics.

Although this is a quite interesting issue within the scope of conservation biology, there is a huge of literature in this topic and so

it is very difficult to understand what is the main contribution of the present work. Additionally, the writing is a bit confusing, and even the modeling is not clear.

My first impression is that there is also some misunderstanding about some of the concepts they talk about.

Below I provide more details about my criticism.

1) As I said before, there is a huge body of literature concerning this problem. What the authors define as an immigration rate could simply be seen as a speciation rate as standard in other models in the literature. On the other hand, the emigration rate, in my point of view, could simply be seen as a death rate. For me, it is a bit difficult to understand the real difference between the two approaches.

2) On page 2, the authors state: "The immigration rate is v necessary for the model; otherwise, no species will exist in the study area, and community dynamics are not possible if the area originally had no organisms in existence, and no individuals could disperse into the area from the outside". As far as know, if the speciation rate is null, then if one has a contiguous area, the number of species goes to one in the long term. In case one has a fragmented landscape, the biodiversity can be higher. Why can no species survive if there is no immigration?

3) The set of equations (1) is presented, but the quantities are not defined. For example, what is $p(j,t)$? I suppose it is the probability a given species has abundance j at time t . But this information should be provided. Please also explain each term in the right of the equation.

4) Looking at the solution of the probability generating function in Eq. (4), I see that for $z=1$ (the case $u=0$) $H(z,t)$ becomes undefined. In fact for $z=1$ one should obtain $H(1,t)=1$.

5) The authors also state that the initial condition is $p(n,0 | j) = \delta(n-j)$, and so now one has an initial condition for a conditional probability and not for $p(j,t)$. In Eq. (5), the authors tell about probability moments for $p(n,0 | j)$???

6) The definition of $S(0 | A)$, $N(0 | A)$ and $S_{\{j\}}(0 | A)$ is also puzzling. My impression is that $N(0 | A)$ is not necessary.

7) On page 10, it is written: "As a consequence, our study actually already quantified temporal behaviors of extinction debt over time, which is simply the expected species richness over time like Eq. 12 (as a demonstration, please refer to Fig. S2)."

My impression is that the authors misunderstood the meaning of extinction debt. Extinction debt is an estimate for the number of endangered species.

Review form: Reviewer 2

Is the manuscript scientifically sound in its present form?

No

Are the interpretations and conclusions justified by the results?

No

Is the language acceptable?

No

Do you have any ethical concerns with this paper?

No

Have you any concerns about statistical analyses in this paper?

No

Recommendation?

Major revision is needed (please make suggestions in comments)

Comments to the Author(s)

See attached PDF (Appendix A).

Decision letter (RSOS-191039.R0)

20-Aug-2019

Dear Professor Shen,

The editors assigned to your paper ("Extinction accumulation in local habitats: quantifying the roles of random drift, immigration, and emigration") have now received comments from reviewers. We would like you to revise your paper in accordance with the referee and Associate Editor suggestions which can be found below (not including confidential reports to the Editor). Please note this decision does not guarantee eventual acceptance.

Please submit a copy of your revised paper before 12-Sep-2019. Please note that the revision deadline will expire at 00.00am on this date. If we do not hear from you within this time then it will be assumed that the paper has been withdrawn. In exceptional circumstances, extensions may be possible if agreed with the Editorial Office in advance. We do not allow multiple rounds of revision so we urge you to make every effort to fully address all of the comments at this stage. If deemed necessary by the Editors, your manuscript will be sent back to one or more of the original reviewers for assessment. If the original reviewers are not available, we may invite new reviewers.

- Data accessibility

If you wish to submit your supporting data or code to Dryad (<http://datadryad.org/>), or modify your current submission to dryad, please use the following link:
<http://datadryad.org/submit?journalID=RSOS&manu=RSOS-191039>

- Competing interests

- Authors' contributions

- Acknowledgements

- Funding statement

Kind regards,

Alice Power

Editorial Coordinator

on behalf of Dr Punidan Jeyasingh (Associate Editor) and Kevin Padian (Subject Editor)

Associate Editor's comments (Dr Punidan Jeyasingh):

This paper takes a modeling approach to address key questions in evolutionary ecology. The topic is timely, and of general interest. The manuscript was reviewed by two experts, both of whom saw merit in the topic and work. Nevertheless, both have raised several criticisms about the work. I felt the reviews were clear, fair, and constructive. I'd like to give the authors a chance to respond to these issues, if they can. I also thank the expert reviewers for their thoughtful comments.

Subject Editor Comments to Author:

The reviewers note some major problems with the equations and the presentation of the paper, but they also provide suggested corrections that should address these problems. Please attend carefully to these individual comments, and if you need more time to revise, contact the editorial office. Best wishes for your revision.

Reviewers' Comments to Author:

Reviewer: 1

Comments to the Author(s)

In this submission, Chen et al. present a stochastic neutral model to estimate the biodiversity change under ecological disturbance and dispersal dynamics. Although this is a quite interesting issue within the scope of conservation biology, there is a huge of literature in this topic and so it is very difficult to understand what is the main contribution of the present work. Additionally, the writing is a bit confusing, and even the modeling is not clear. My first impression is that there is also some misunderstanding about some of the concepts they talk about. Below I provide more details about my criticism.

1) As I said before, there is a huge body of literature concerning this problem. What the authors define as an immigration rate could simply be seen as a speciation rate as standard in other models in the literature. On the other hand, the emigration rate, in my point of view, could

simply be seen as a death rate. For me, it is a bit difficult to understand the real difference between the two approaches.

2) On page 2, the authors state: "The immigration rate is unnecessary for the model; otherwise, no species will exist in the study area, and community dynamics are not possible if the area originally had no organisms in existence, and no individuals could disperse into the area from the outside". As far as know, if the speciation rate is null, then if one has a contiguous area, the number of species goes to one in the long term. In case one has a fragmented landscape, the biodiversity can be higher. Why can no species survive if there is no immigration?

3) The set of equations (1) is presented, but the quantities are not defined. For example, what is $p(j,t)$? I suppose it is the probability a given species has abundance j at time t . But this information should be provided. Please also explain each term in the right of the equation.

4) Looking at the solution of the probability generating function in Eq. (4), I see that for $z=1$ (the case $u=0$) $H(z,t)$ becomes undefined. In fact for $z=1$ one should obtain $H(1,t)=1$.

5) The authors also state that the initial condition is $p(n,0 | j) = \delta(n-j)$, and so now one has an initial condition for a conditional probability and not for $p(j,t)$. In Eq. (5), the authors tell about probability moments for $p(n,0 | j)$???

6) The definition of $S(0 | A)$, $N(0 | A)$ and $S_{\{j\}}(0 | A)$ is also puzzling. My impression is that $N(0 | A)$ is not necessary.

7) On page 10, it is written: "As a consequence, our study actually already quantified temporal behaviors of extinction debt over time, which is simply the expected species richness over time like Eq. 12 (as a demonstration, please refer to Fig. S2)." My impression is that the authors misunderstood the meaning of extinction debt. Extinction debt is an estimate for the number of endangered species.

Reviewer: 2

Comments to the Author(s)
See attached PDF

Author's Response to Decision Letter for (RSOS-191039.R0)

Appendix B.

RSOS-191039.R1 (Revision)

Review form: Reviewer 1

Is the manuscript scientifically sound in its present form?

Yes

Are the interpretations and conclusions justified by the results?

Yes

Is the language acceptable?

Yes

Is it clear how to access all supporting data?

N/A

Do you have any ethical concerns with this paper?

No

Have you any concerns about statistical analyses in this paper?

No

Recommendation?

Accept with minor revision (please list in comments)

Comments to the Author(s)

I think the authors have done a great effort to improve the manuscript and solve the critical issues, as raised in my previous reports. Before the acceptance of the paper, the authors should address some minor points:

1) The y-axis label must be changed in the figures. In fact, the correct is "number of species" (or biodiversity level) instead of total species loss. The species loss corresponds to the gap between curves before and after disturbance.

2) The authors should compare their outcomes with those already shown in the literature. As examples of studies on this issue please see Claudino et al., Extinction debt and the role of static and dynamical fragmentation on biodiversity. *Ecological Complexity* (Print), v. 21, p. 150-155, 2015; Gomes et al. Effect of Landscape Structure on Species Diversity. *Plos One*, v. 8, p. e66495, 2013; Thompson et al., Characterising extinction debt following habitat fragmentation using neutral theory, *Ecol. Letters* (2019); Coelho Neto et al., Neutral communities in fragmented landscapes. *Oikos* (Kobenhavn), v. 121, p. 1737-1748, 2012.

Review form: Reviewer 2

Is the manuscript scientifically sound in its present form?

Yes

Are the interpretations and conclusions justified by the results?

Yes

Is the language acceptable?

Yes

Is it clear how to access all supporting data?

Yes

Do you have any ethical concerns with this paper?

No

Have you any concerns about statistical analyses in this paper?

No

Recommendation?

Accept with minor revision (please list in comments)

Comments to the Author(s)

[REVISED PAPER] Journal, Manuscript ID: Royal Society Open Science - RSOS-191039

By Y Chen, S-C Chang, Y Wu, Y-F C, T-J Shen

The authors have addressed all of my concerns seriously and carefully. Only one thing remains, though. Regarding my 9th point: "The 'convex trend' in Figs 1-3 prior to habitat loss, is probably an increase in diversity due to starting diversity at a low value (smaller than equilibrium). The authors should allow the system time to reach equilibrium prior to the habitat destruction. An extra figure might explore the convex or concave trends for different starting values of biodiversity.'

To this the authors reply "Actually, the convex or concave trend is dependent on the time-dependent solution of the extinction curve, but not the initial community configuration. That is, regardless of how species abundance distribution is for the initial community ($t = 0$), the curve shape (convex versus concave) is totally determined by the time-dependent solution of the net biodiversity loss (i.e., species richness at time t versus that at time zero) ... One can see is always $S'(t)$ negative; meanwhile, $S'(t)$ is increasing with t . To this end, it can be concluded that Sgardeli et al. (2017)'s main equation is convex."

But wait. The time-dependent solution of the extinction curve contains S_0 , the initial species richness. Note that $S'(t)$ will have a different sign according to the factor ($S_0 - S_{eq}$). Thus $S'(t)$ can be positive or negative according to the initial diversity S_0 . This makes sense. If we start above equilibrium, diversity must diminish and if we start below it must increase.

But my point is that Fig1 as it stands is confusing because we need more ecological context. Thus,... insert a second panel into Fig 1 (not an Appendix) to show $S(t)$ as a function of time, and the problem is solved. This will help the readers greatly.

Decision letter (RSOS-191039.R1)

11-Nov-2019

Dear Professor Shen,

On behalf of the Editors, I am pleased to inform you that your Manuscript RSOS-191039.R1 entitled "Extinction debt in local habitats: quantifying the roles of random drift, immigration, and emigration" has been accepted for publication in Royal Society Open Science subject to minor revision in accordance with the referee suggestions. Please find the referees' comments at the end of this email.

The reviewers and Subject Editor have recommended publication, but also suggest some minor revisions to your manuscript. Therefore, I invite you to respond to the comments and revise your manuscript.

- Ethics statement

- Data accessibility

<http://datadryad.org/submit?journalID=RSOS&manu=RSOS-191039.R1>

- Competing interests

- Authors' contributions

- Acknowledgements

- Funding statement

Because the schedule for publication is very tight, it is a condition of publication that you submit the revised version of your manuscript before 20-Nov-2019. Please note that the revision deadline will expire at 00.00am on this date. If you do not think you will be able to meet this date please let me know immediately.

Kind regards,
Lianne Parkhouse

Editorial Coordinator
 Royal Society Open Science
 openscience@royalsociety.org

on behalf of Dr Punidan Jeyasingh (Associate Editor) and Kevin Padian (Subject Editor)
 openscience@royalsociety.org

Associate Editor Comments to Author (Dr Punidan Jeyasingh):

I thank the authors for a thorough revision of the original manuscript based upon highly constructive expert reviewer comments. This version was reassessed by these experts, both of whom were largely satisfied with the revisions. Nevertheless, they have raised a few issues that I felt were fair and constructive. I invite the authors to make these final adjustments, and look forward to seeing this paper in print.

Reviewer comments to Author:

Reviewer: 1
 Comments to the Author(s)

I think the authors have done a great effort to improve the manuscript and solve the critical issues, as raised in my previous reports. Before the acceptance of the paper, the authors should address some minor points:

1) The y-axis label must be changed in the figures. In fact, the correct is "number of species" (or biodiversity level) instead of total species loss. The species loss corresponds to the gap between curves before and after disturbance.

2) The authors should compare their outcomes with those already shown in the literature. As examples of studies on this issue please see Claudino et al., Extinction debt and the role of static and dynamical fragmentation on biodiversity. *Ecological Complexity* (Print), v. 21, p. 150-155, 2015; Gomes et al. Effect of Landscape Structure on Species Diversity. *Plos One*, v. 8, p. e66495, 2013; Thompson et al., Characterising extinction debt following habitat fragmentation using neutral theory, *Ecol. Letters* (2019); Coelho Neto et al., Neutral communities in fragmented landscapes. *Oikos* (Kobenhavn), v. 121, p. 1737-1748, 2012.

Reviewer: 2
 Comments to the Author(s)

[REVISED PAPER] Journal, Manuscript ID: Royal Society Open Science - RSOS-191039
 By Y Chen, S-C Chang, Y Wu, Y-F C, T-J Shen

The authors have addressed all of my concerns seriously and carefully. Only one thing remains, though. Regarding my 9th point: "The 'convex trend' in Figs 1-3 prior to habitat loss, is probably an increase in diversity due to starting diversity at a low value (smaller than equilibrium). The authors should allow the system time to reach equilibrium prior to the habitat destruction. An extra figure might explore the convex or concave trends for different starting values of biodiversity.'

To this the authors reply “Actually, the convex or concave trend is dependent on the time-dependent solution of the extinction curve, but not the initial community configuration. That is, regardless of how species abundance distribution is for the initial community ($t = 0$), the curve shape (convex versus concave) is totally determined by the time-dependent solution of the net biodiversity loss (i.e., species richness at time t versus that at time zero) ... One can see is always $S'(t)$ negative; meanwhile, $S'(t)$ is increasing with t . To this end, it can be concluded that Sgardeli et al. (2017)'s main equation is convex.”

But wait. The time-dependent solution of the extinction curve contains S_0 , the initial species richness. Note that $S'(t)$ will have a different sign according to the factor $(S_0 - S_{eq})$. Thus $S'(t)$ can be positive or negative according to the initial diversity S_0 . This makes sense. If we start above equilibrium, diversity must diminish and if we start below it must increase.

But my point is that Fig1 as it stands is confusing because we need more ecological context. Thus,... insert a second panel into Fig 1 (not an Appendix) to show $S(t)$ as a function of time, and the problem is solved. This will help the readers greatly.

Author's Response to Decision Letter for (RSOS-191039.R1)

See Appendix C.

Decision letter (RSOS-191039.R2)

28-Nov-2019

Dear Professor Shen,

It is a pleasure to accept your manuscript entitled "Extinction debt in local habitats: quantifying the roles of random drift, immigration, and emigration" in its current form for publication in Royal Society Open Science. The comments of the reviewer(s) who reviewed your manuscript are included at the foot of this letter.

Due to rapid publication and an extremely tight schedule, if comments are not received, your

paper may experience a delay in publication. Royal Society Open Science operates under a continuous publication model. Your article will be published straight into the next open issue and this will be the final version of the paper. As such, it can be cited immediately by other researchers. As the issue version of your paper will be the only version to be published I would advise you to check your proofs thoroughly as changes cannot be made once the paper is published.

on behalf of Dr Punidan Jeyasingh (Associate Editor) and Kevin Padian (Subject Editor)
openscience@royalsociety.org

Follow Royal Society Publishing on WeChat
Follow Royal Society Publishing on Twitter: @RSocPublishing
Follow Royal Society Publishing on Facebook:
<https://www.facebook.com/RoyalSocietyPublishing.FanPage/>
Read Royal Society Publishing's blog: <https://blogs.royalsociety.org/publishing/>

Appendix A

Extinction accumulation in local habitats: quantifying the roles of random drift, immigration, and emigration

Journal, Manuscript ID: Royal Society Open Science - RSOS-191039

By Y Chen, S-C Chang, Y Wu, Y-F C, T-J Shen

The authors use a birth-and-death stochastic model to derive results relevant to extinction debt. I am very fond of this kind of analysis in ecology and thus encourage the authors in this work. The paper is very close in its result to Gilbert et al 2006 but modifies that result into something different. In Gilbert et al, all extinctions are due to random drift towards zero but here the authors consider random drift plus immigration-emigration. The paper claims to deliver a number of nice results and may deliver them, eventually. Unfortunately, right now I am unable to recommend publication of this paper in its current state. A corrected paper may be publishable.

Main issues

1. The description of the ecological model behind the birth-death equations (1) is totally inadequate. The authors should state the ecological nature of the model they assume, such as
 - Single species
 - single location
 - no intrinsic growth rate (?)
 - no density dependence
 - no interaction with other species
 - ... ?

In this way the reader can situate themselves and see what is new

2. Make Eq.(1) itself clearer. Everything hangs on Eq. (1). The current form is opaque and needs simplification. I would suggest the authors use terminology that is more standard. My reworking looks like this (the 't' argument is not needed here as all p's are time dependent):

$$\frac{dp_0}{dt} = -vp_0 + p_1$$

$$\frac{dp_1}{dt} = +vp_0 - (2-u)p_1 + 2p_2$$

...

$$\frac{dp_j}{dt} = +(j-1)(1-u)p_{j-1} - j(2-u)p_j + (j+1)(1-u)p_{j+1}$$

A figure might also help. The Wiki article (Birth-death_process) is a decent example.

3. Something is wrong with Eq. (6) and following if immigration rate is not relevant to survival at time t . This cannot be right, I think. After all, if we start off with a population of zero, then for immigration to rescue this, we have to wait. Thus the lower that value of immigration, the lower the probability of a nonzero p_j at time t . Compare with Sgardeli et al's formula for equilibrium where both deathrate and speciation play a role. Check this.

4. Something else seems wrong with Eq. (1). Suppose the population starts with 3 individuals so $p_j(0)=\delta_{3j}$. Then there is a probability that this could rise to 4 in the next time-step.

$$\frac{dp_4}{dt} = +3(1-u)p_3(0)$$

However, there could also happen an immigration event here. So, there should also be some dependence on v in the higher states and not only in $j=0$ or $j=1$. Check this too.

5. The paper is spoiled by poor description and terminology. In general, the authors use terminology that is unusual for the field and which is therefore confusing. In particular, I strongly encourage the authors do avoid using the term *cumulative extinctions*, since this could mean either biodiversity loss by time $t=t$ relative to time $t=0$ or else the total number of extinctions that have happened in this interval. For example, a community might lose no species overall but one species might have gone extinct and been rescued 3 times. So *cumulative extinctions* could number either 0 or 3.
6. Following the preceding, I suggest the title be “Extinction debt in local habitats: quantifying the roles of random drift, immigration, and emigration”
7. The authors must tell readers what the parameters mean. This manuscript contains numerous equations without description of parameters. An example (and there are many more!!) is on page 5, line 29: the equation for the truncated negative binomial.
8. Overall, the coverage of current literature is good. But the authors should also compare their own results with those Sgardeli et al 2017 which also considers extinction debt subject to speciation and also the possibility of using speciation as a proxy for immigration-emigration.
9. The “convex trend” in Figs 1-3 prior to habitat loss, is probably an increase in diversity due to starting diversity at a low value (smaller than equilibrium). The authors should allow the system time to reach equilibrium prior to the habitat destruction. An extra figure might explore the convex or concave trends for different starting values of biodiversity.

References

1. Sgardeli V, Iwasa Y, Varvoglis H, Halley JM. A forecast for extinction debt in the presence of speciation. *Journal of theoretical biology*. 2017; 415:48-52.

Appendix B

Response Letter

Dear Editors,

Thanks for providing us with an opportunity to improve our paper according to the constructive comments of the handling editors and two reviewers. We really value the hard-won revision chance and tried our best efforts in revising the manuscript after receiving the report. The point-to-point responses for all comments proposed by the handling editors and the reviewers have been presented in *blue* color in this letter.

Moreover, for your perusal, the changes or revisions made in the main text and the appendix have been highlighted in *red* color. Finally, to make our proposed method become reproducible, we present the R code for the simulation of representative figures presented in the paper and has been deposited in Dryad database (<https://datadryad.org/review?doi=doi:10.5061/dryad.n710qb3>).

We sincerely hope that our substantial revision could significantly improve the paper to reach the high-publishing standard of your journal, and look forward to hearing back from you.

Yours Sincerely,
The authors

Dear Professor Shen,

The editors assigned to your paper ("Extinction accumulation in local habitats: quantifying the roles of random drift, immigration, and emigration") have now received comments from reviewers. We would like you to revise your paper in accordance with the referee and Associate Editor suggestions which can be found below (not including confidential reports to the Editor). Please note this decision does not guarantee eventual acceptance.

Please submit a copy of your revised paper before 12-Sep-2019. Please note that the revision deadline will expire at 00.00am on this date. If we do not hear from you within this time then it will be assumed that the paper has been withdrawn. In exceptional circumstances, extensions may be possible if agreed with the Editorial Office in advance. We do not allow multiple rounds of revision so we urge you to make every effort to fully address all of the comments at this stage. If deemed necessary by the Editors, your manuscript will be sent back to one or more of the original reviewers for assessment. If the original reviewers are not available, we may invite new reviewers.

Response: this ethnic statement is not applicable to our paper.

- Data accessibility

Response: provided.

<http://datadryad.org/submit?journalID=RSOS&manu=RSOS-191039>

Response: we have updated a Dryad repository record for the revised paper as

[https://datadryad.org/review?doi=doi:10.5061/dryad.n710qb3.](https://datadryad.org/review?doi=doi:10.5061/dryad.n710qb3)

- Competing interests

Response: a competing interest statement is provided for the revised paper.

- Authors' contributions

AB carried out the molecular lab work, participated in data analysis, carried out sequence alignments, participated in the design of the study and drafted the manuscript; CD carried out the

statistical analyses; EF collected field data; GH conceived of the study, designed the study, coordinated the study and helped draft the manuscript. All authors gave final approval for publication.

Response: a section of author contributions is provided for the revised paper by following your suggestion.

- Acknowledgements

Response: provided.

- Funding statement

Response: a funding statement is provided.

Kind regards,

Alice Power

Editorial Coordinator

on behalf of Dr Punidan Jeyasingh (Associate Editor) and Kevin Padian (Subject Editor)

Associate Editor's comments (Dr Punidan Jeyasingh):

This paper takes a modeling approach to address key questions in evolutionary ecology. The topic is timely, and of general interest. The manuscript was reviewed by two experts, both of whom saw merit in the topic and work. Nevertheless, both have raised several criticisms about the work. I felt the reviews were clear, fair, and constructive. I'd like to give the authors a chance to respond to these issues, if they can. I also thank the expert reviewers for their thoughtful comments.

Response: thank you and the expert reviewers for providing us with a hard-won chance to revise our paper. We have tried our best to revise the paper according to each valuable comment of the decision letter. After following your constructive suggestions, we believe the revised manuscript is more suited for potential publication in the journal than our earlier version.

Subject Editor Comments to Author:

The reviewers note some major problems with the equations and the presentation of the paper, but they also provide suggested corrections that should address these problems. Please attend carefully to these individual comments, and if you need more time to revise, contact the editorial office.

Best wishes for your revision.

Response: thank you for providing us with the opportunity to improve our paper. We have tried our best to revise the paper according to each comment of the decision letter. We have considerably and carefully revised the manuscript which should be more readable and easier to follow than the preceding version; please refer to our detailed responses to all comments and suggestions below.

Reviewers' Comments to Author:

Reviewer: 1

Comments to the Author(s)

In this submission, Chen et al. present a stochastic neutral model to estimate the biodiversity change under ecological disturbance and dispersal dynamics. Although this is a quite interesting issue within the scope of conservation biology, there is a huge of literature in this topic and so it is very difficult to understand what is the main contribution of the present work. Additionally, the writing is a bit confusing, and even the modeling is not clear. My first impression is that there is also some misunderstanding about some of the concepts they talk about. Below I provide more details about my criticism.

Response: thanks for the critical evaluation on the paper and pointed out the ambiguities about the concepts in the paper. We have tried our best to revise the paper following your constructive comments. We believe the novelty of our paper is that we may be the first one to provide a time-dependent model that can describe both extinction debt and immigration credits. By contrast, previous studies only discussed these terms but never observe the temporal trajectory of immigration credits. However, the model proposed in our paper can additionally describe the emergence of immigration credits.

1) As I said before, there is a huge body of literature concerning this problem. What the authors define as an immigration rate could simply be seen as a speciation rate as standard in other models in the literature. On the other hand, the emigration rate, in my point of view, could simply be seen as a death rate. For me, it is a bit difficult to understand the real difference between the two approaches.

Response: we agreed with your standpoint. However, the reason that we define immigration and emigration rates instead of birth and death ones by simply following previous literature (He 2005, Volkov et al. 2005, Allen 2010). More importantly, the birth and death rates actually have been implicitly taken into account in our model (Eq. 1 of the main text), both of which are set to 1; please refer to a new Table 1 of the revised main text for the discussion. Additionally, the framework of this paper allows us to extensively discuss effects of immigration and emigration on species loss (please refer to the updated Figs. 3-4 in the revised main text and Fig. S3 in the updated appendix).

2) On page 2, the authors state: "The immigration rate is v unnecessary for the model; otherwise, no species will exist in the study area, and community dynamics are not possible if the area originally had no organisms in existence, and no individuals could disperse into the area from the outside". As far as know, if the speciation rate is null, then if one has a contiguous area, the number of species goes to one in the long term. In case one has a fragmented landscape, the biodiversity can be higher. Why can no species survive if there is no immigration?

Response: we agree with your viewpoint based on the classical zero-sum competition neutral theory, in which, no speciation rate is involved; in a long term, only a single species persists in the community. However, our model (Eq. 1 in the main text) allows immigrants of the same species to disperse into the target area, allowing species to persist (therefore, it becomes less likely to go extinct). Moreover, our model is a single species-specific dynamic model (see Table 1 of the revised main text for the reference), which does not assume the competition of different species. To this end, the zero-sum game observed in the classical competitive neutral model is not applicable to our model.

Your specific comment here is also similar to one comment mentioned by Reviewer 2. Actually, our model (Eq. 1) has a strong assumption: that is, an immigration event can only take place when there are no individuals in the target area. When there are some individuals occurring in the target area, our neutral model (Eq. 1) assumes that the immigration rate with respect to intrinsic growth or death rate is negligible (therefore, immigration rate did not occur in the higher-abundance transition Eq. 1 for $j > 1$). A partial reason of our assumption can come from that potential immigrants may not have an appealing boost to move into the target area when there are some individuals since they need to undergo a high competition pressure.

3) The set of equations (1) is presented, but the quantities are not defined. For example, what is $p(j,t)$? I suppose it is the probability a given species has abundance j at time t . But this information should be provided. Please also explain each term in the right of the equation.

Response: thanks for the comment. Yes, $p(j, t)$ (in the revised text, following the suggestion of Reviewer 2, we change the annotation as $p_j(t)$) is the probability that a given species has abundance j at time t . We have added the definition of $p_j(t)$ right after Eq. 1 of the revision.

To further clarify the right-hand side of Eq. 1 along with a similar suggestion by Reviewer 2, we create a flowchart to illustrate them in detail, please see Fig. S1 of the revised appendix of this revision. Moreover, to follow your suggestion and Reviewer 2, we created a new table (Table 1 in the revised main text) in the revised text for explaining the model assumption, each mathematical symbol and parameter value assignment for the stochastic model in Eq. 1. We believe these efforts can make readers better understand and be familiar with the model used in our paper.

4) Looking at the solution of the probability generating function in Eq. (4), I see that for $z=1$ (the case $u=0$) $H(z,t)$ becomes undefined. In fact for $z=1$ one should obtain $H(1,t)=1$.

Response: thanks for pointing out this issue to us. We have carefully re-checked our solution of the proposed model and found a critical problem on the solution. We found the current solution (probability generating function) as Eq. 4 in the original paper was incorrect. After consulting a theoretical physicist (Dr. Qingdi Wang from University of British Columbia) and with our own derivation, we confirm that, when the immigration rate v is not zero, there is no analytical solution for the PDE of the probability generating function (Eq. 3 in the original paper), as $p_0(t)$ has no

analytical formulae in the model. When the immigration rate ν is zero, $H(z,t)$ can be found in Rosindell et al. (2010)'s paper and also presented in the revised paper (please refer to the new Eqs. 2a and 2b of the revision).

When the immigration rate ν is not zero (which is a focus in our paper), we decide to utilize numerical approaches to resolve our stochastic model (Eq. 1) using the matrix exponential algorithm (please refer to new Eqs. 4-5 in the revised paper about the numerical computing method).

5) The authors also state that the initial condition is $p(n,0|j)=\delta(n-j)$, and so now one has an initial condition for a conditional probability and not for $p(j,t)$. In Eq. (5), the authors tell about probability moments for $p(n,0|j)$???

Response: $p(n,0|j)$ (in the revised text, the mathematical annotation is changed to $p_n(0)$) is determined by the initial abundance condition $\delta(n-j)$. That is, when a species has an

abundance of j at the initial condition, the initial probability is $\begin{cases} p_n(0) = 1, & \text{if } n = j; \\ p_n(0) = 0, & \text{if } n \neq j. \end{cases}$ This is the

meaning of $\delta(n-j)$; we have given the specific definition about the mathematical function δ right after Eq. 2b.

Sorry, in revision, we have deleted the typo "moments" from the sentence you mentioned.

6) The definition of $S(0|A)$, $N(0|A)$ and $S_{\{j\}}(0|A)$ is also puzzling. My impression is that $N(0|A)$ is not necessary.

Response: $N(0|A)$ and $S(0|A)$ represent the initial community configuration for the total abundance and species richness, respectively, they can determine the changing curvature of species richness or extinction over time in the model. Such an initial community configuration along with its influence on community dynamic under a different neutral model is also presented in a dynamic model in Sgardeli et al. (2017) (Fig. 1 in their paper) and other static (or non-dynamic) models of extinction debts (Kitzes and Harte 2015, Chen and Shen 2017). However, we agree that $N(0|A)$ is unnecessarily be mentioned in our paper and has been deleted.

7) On page 10, it is written: "As a consequence, our study actually already quantified temporal behaviors of extinction debt over time, which is simply the expected species richness over time like Eq. 12 (as a demonstration, please refer to Fig. S2)." My impression is that the authors misunderstood the meaning of extinction debt. Extinction debt is an estimate for the number of endangered species.

Response: in fact, the concept of extinction debt is tightly related to time. It is a function of time and extinction debt will change before reaching equilibrium. That is, it is safe to say that extinction debt is a temporal phenomenon. As an example, Halley et al. (2016) also investigated the temporal dynamic of extinction debt.

Comments from Reviewer 2:

Extinction accumulation in local habitats: quantifying the roles of random drift, immigration, and emigration

Journal, Manuscript ID: Royal Society Open Science - RSOS-191039

By Y Chen, S-C Chang, Y Wu, Y-F C, T-J Shen

The authors use a birth-and-death stochastic model to derive results relevant to extinction debt. I am very fond of this kind of analysis in ecology and thus encourage the authors in this work. The paper is very close in its result to Gilbert et al 2006 but modifies that result into something different. In Gilbert et al, all extinctions are due to random drift towards zero but here the authors consider random drift plus immigration-emigration. The paper claims to deliver a number of nice results and may deliver them, eventually. Unfortunately, right now I am unable to recommend publication of this paper in its current state. A corrected paper may be publishable.

Response: thanks for your precious time on reading our paper along with insightful comments. We have tried our best to revise the paper according to your comments; please see our detailed responses below.

Main issues

1. The description of the ecological model behind the birth-death equations (1) is totally inadequate. The authors should state the ecological nature of the model they assume, such as

- Single species
- single location
- no intrinsic growth rate (?)
- no density dependence
- no interaction with other species
- ... ?

In this way the reader can situate themselves and see what is new

Response: thanks for the suggestion. In the revision, we created a brand-new table (Table 1 in the revised main text) about a list of assumptions and parameter configurations for the model used in the present study.

2. Make Eq.(1) itself clearer. Everything hangs on Eq. (1). The current form is opaque and needs simplification. I would suggest the authors use terminology that is more standard. My reworking looks like this (the 't' argument is not needed here as all p's are time dependent):

$$\frac{dp_0}{dt} = -vp_0 + p_1$$

$$\frac{dp_1}{dt} = +vp_0 - (2-u)p_1 + 2p_2$$

...

$$\frac{dp_j}{dt} = +(j-1)(1-u)p_{j-1} - j(2-u)p_j + (j+1)(1-u)p_{j+1}$$

A figure might also help. The Wiki article (Birth-death_process) is a decent example.

Response: thanks for the good idea, following your suggestion, we used the simplified version with replacing \$p(j, t)\$ by \$p_j(t)\$ in Eq. 1 of the revision. However, we still keep the time annotation \$t\$ in the transition probabilities \$p_j(t)\$. This is because, we have to extensively discuss

the influence of initial condition $t=0$ on the transition probabilities in subsequent modeling (for example, Eq. 5 in the main text and Fig. S2 of the Appendix). Moreover, the new annotation is also consistent with richness and abundance annotations for the initial community configuration in the paper (e.g., $S(0|A)$ and $S_j(0|A)$).

Finally, as suggested, a flow chat visually illustrating the transitional relationship between different abundance states (j) is provided as the new Fig. S1A of the Appendix (the order of other supplemental figures is updated accordingly). Moreover, the flowchart for the transition dynamic between neighboring states for a full immigration model (newly considered in the revision) is also presented as Fig. S1B of the Appendix.

3. Something is wrong with Eq. (6) and following if immigration rate is not relevant to survival at time t . This cannot be right, I think. After all, if we start off with a population of zero, then for immigration to rescue this, we have to wait. Thus the lower that value of immigration, the lower the probability of a nonzero p_j at time t . Compare with Sgardeli et al's formula for equilibrium where both deathrate and speciation play a role. Check this.

Response: thanks for the suggestion and pointing out the issue to us. We have carefully re-checked our solution of the proposed model and found a critical problem on the solution. We found the current solution (probability generating function) as Eq. 4 in the original paper was incorrect. After consulting a theoretical physicist (Dr. Qingdi Wang from University of British Columbia) and with our own derivation, we confirm that, when immigration rate ν is nonzero, there is no analytical solution for the PDE of the probability generating function (Eq. 3 in the original paper), as $H(0|t)$ in Eq. 2a of the revision has no analytical formulae in the model.

Therefore, we decide to employ numerical approaches to resolve stochastic model (Eq. 1) using the matrix exponential algorithm (please refer to new Eqs. 4-5 in the revised paper). Further, we guarantee that the numerical calculation of $p_0(t)$ in the revised paper is correct, as we verified its calculation independently by different co-authors of the paper. The numerical $p_0(t)$ values are clearly related to both immigration and emigration rates now, as shown below (this figure is also presented as Fig. S3 in Supporting Information of the revised paper):

From this figure, one can see, when the immigration rate is high while the emigration rate is low, the probability of extinction $p_0(t)$ is low. By contrast, when the immigration rate is low while the emigration rate is high, the probability of extinction $p_0(t)$ becomes high and reaches 1 when time is sufficiently large.

Moreover, thanks for introducing Sgardeli et al. (2017)'s paper to us, formulae in which indeed are very helpful and are derived from a neutral model at the community level. By contrast, the model used in our paper is a single-species model (see the new Table 1 of the revised paper for discussion). In the revised paper, we compared our model (Eq. 7) to Sgardeli et al. (2017)'s formula on time-dependent species richness as

$$S(t) = S_{eq} + \frac{2S_{eq}}{\frac{S_0 + S_{eq}}{S_0 - S_{eq}} e^{rt} - 1},$$

where $S_{eq} = J \sqrt{\frac{w}{w+b}}$, J is the community size, S_0 is the initial community species richness, w is the speciation rate, and u is the birth rate. Moreover, $r = 2w/S_{eq}$ represents the relaxation rate in Sgardeli et al. (2017)'s paper. This is a concave function; please see our responses to your other comments below. However, the species loss model can be formulated as

$$E_S(t) = S_0 - S(t) = S_0 - S_{eq} - \frac{2S_{eq}}{\frac{S_0 + S_{eq}}{S_0 - S_{eq}} e^{rt} - 1}.$$

Apparently, $E_S(t)$ becomes convex. In our paper, at the metacommunity level, the general species loss model is presented in Eq. 8 in the revised paper as

$$E(t | A) = S(0 | A) - S(t | A).$$

When the immigration rate does not exist (i.e., $v=0$), we can write its analytical form as follows:

$$E(t | A) = \begin{cases} \sum_{j=1}^{\infty} S_j(0 | A) \left(\frac{t}{1+t} \right)^j, & u = 0; \\ \sum_{j=1}^{\infty} S_j(0 | A) \left(\frac{1 - e^{-ut}}{1 - (1-u)e^{-ut}} \right)^j, & u > 0. \end{cases}$$

However, when immigration rate is not negligible, there is no analytical form for $E(t|A)$, and thus it will be resolved out numerically.

4. Something else seems wrong with Eq. (1). Suppose the population starts with 3 individuals so $p_j(0)=\delta_{3j}$. Then there is a probability that this could rise to 4 in the next time-step.

$$\frac{dp_4}{dt} = +3(1-u)p_3(0)$$

However, there could also happen an immigration event here. So, there should also be some dependence on v in the higher states and not only in $j=0$ or $j=1$. Check this too.

Response: we agree that some dependence on v in higher states ($j>1$) might be more realistic. However, at the meanwhile, we assume that the arrival of a new immigrant is only possible when there are no individuals of the same species in the target community. This means, we assume that strict competitive exclusion effect exists between native individuals and immigrants, and the immigration event is possible when there are no native conspecifics in the community. However, native conspecifics (i.e., those individuals that are created by birth events) can coexist without exclusion.

This assumption seems very strong but is still realistic and a bit similar to the rescue effect in the ecological literature (Brown and Kodric-Brown 1977, Gotelli 1991, Stacey and Taper 1992, Dornier and Cheptou 2012, Lawson et al. 2012, Eriksson et al. 2014). In our paper, the rescue effect takes place through immigration from other areas only when the population size of a focal species in a target area is sufficiently small. That is, the rescue effect (v) is magnified at most when there is no individuals of the target species in the target area. By contrast, the rescue effect (v) is ignored in our model (Eq. 1 in the main text) when there are some individuals of the target species in the focused area. This is because the population is dynamically dominated by both density-dependent growth and death rates (both rates=1, see Table 1 in the revised main text for explanation). The density-independent immigration rescue effect is therefore negligible (as v is usually small, see Table 1 in the revised main text for explanation).

Previous studies (Stacey and Taper 1992, Dornier and Cheptou 2012, Lawson et al. 2012, Eriksson et al. 2014) working on the rescue effect also stress that importance of the rescue effect for species that are extinction-prone and in small population sizes, akin to our model structure and assumption (but the difference between theirs and ours is that we only assume the rescue effect or immigration takes place when there are no individuals of the species in the target area).

Additionally, the model structure and assumption implemented in our paper is also a bit similar to the one-migrant-per-generation rule in conservation biology and genetics (Spielman and Frankham 1992, Mills and Allendorf 1996, Hufbauer et al. 2015). In our study, one immigrant (vp_0)

is only allowed for initiating the dynamic of a target population of a species in the studied area.

We have updated relevant discussion in the revised discussion as well to support this rescue effect idea in the stochastic model used (Eq. 1 of our paper).

Finally, to fully address your concern about the immigration dependency at higher abundance states, we additionally introduce and compare a comparable immigration model (Allen 2010), which has a differential equation system as follows:

$$\left\{ \begin{array}{l} \frac{dp_0(t)}{dt} = p_1(t) - vp_0(t) \\ \frac{dp_1(t)}{dt} = vp_0(t) + 2p_2(t) - (2+v)p_1(t) \\ \vdots \\ \frac{dp_j(t)}{dt} = (j+1)p_{j+1}(t) + ((j-1)+v)p_{j-1}(t) - (2j+v)p_j(t) \end{array} \right.$$

Note that this immigration model does not take account of the role of emigration. We compare it with our model (Eq. 1) for simply addressing the higher state-dependent immigration processes.

The probability of extinction or absence, by assuming that both birth and death rates are equal to 1 as the model used in our paper (see Table 1 of the revised text), can be derived as

$$p_0(t) = \frac{t^j}{(1+t)^{j+v}}.$$

As a result, the associated species loss model can be formulated accordingly by

$$E_{new}(t|A) = \sum_{j=1}^{\infty} S_j(0|A)p_0(t) = \sum_{j=1}^{\infty} S_j(0|A) \frac{t^j}{(1+t)^{j+v}}$$

The performance of the species loss model using the new immigration model $E_{new}(t|A)$ actually is quite close to the one used in our paper (Eq. 8 of the revised text), as the curvilinear shapes were nearly matched (please refer to the updated Figs. 1, 2, and 4 in the revised main text). The difference between them is expected and principally comes from the fact that the new model compared here ($E_{new}(t|A)$) does not take the emigration effect into consideration, while our model (Eq. 1 and Eq. 8 of the main text) can incorporate both emigration and immigration effects simultaneously.

5. The paper is spoiled by poor description and terminology. In general, the authors use terminology that is unusual for the field and which is therefore confusing. In particular, I strongly encourage the authors do avoid using the term *cumulative extinctions*, since this could mean either biodiversity loss by time $t=t$ relative to time $t=0$ or else the total number of extinctions that have happened in this interval. For example, a community might lose no species overall but one species might have gone extinct and been rescued 3 times. So *cumulative extinctions* could number either 0 or 3.

Response: sorry for the confusion caused by the terminology used in our original paper. Thanks for the suggestion; in the revised paper, we utilized the term “*total biodiversity loss*” to replace “*cumulative extinctions*” for avoiding the confusions through the entire text. All figure legends in the revised paper have been updated accordingly.

6. Following the preceding, I suggest the title be “Extinction debt in local habitats: quantifying the roles of random drift, immigration, and emigration”

Response: thanks for the specific suggestion for the title. We changed the title as suggested in the revision.

7. The authors must tell readers what the parameters mean. This manuscript contains numerous equations without description of parameters. An example (and there are many more!!) is on page 5, line 29: the equation for the truncated negative binomial.

Response: thanks for pointing out the annotation issue in the original manuscript. In the revision, we tried to provide explanations on each parameter or symbol for the purpose of improving readability and understandability of the audience. Specifically, we have added “ N_A is a random variate to depict the abundance of each species in the entire region A ; k and ω are two positive parameters of the PMF.” right after Eq. (12) in the revised paper to clearly describe all parameters used in the Fisher’s logseries model which you mentioned. In addition to this point, we have updated other places where ambiguities came out.

8. Overall, the coverage of current literature is good. But the authors should also compare their own results with those Sgardeli et al 2017 which also considers extinction debt subject to speciation and also the possibility of using speciation as a proxy for immigration-emigration.

Response: thanks for your suggestion. Now in the revision, we provided an explicit comparison of the revised new model (Eq. 8 in the main text at the metacommunity level) and Sgardeli et al.’s 2017 model (Eq. 4 in their paper), we found that both can return similar values (or at least curve shape pattern) when the relevant parameter values are adequately given (Fig. S4 of the revised appendix).

9. The “convex trend” in Figs 1-3 prior to habitat loss, is probably an increase in diversity due to starting diversity at a low value (smaller than equilibrium). The authors should allow the system time to reach equilibrium prior to the habitat destruction. An extra figure might explore the convex or concave trends for different starting values of biodiversity.

Response: thanks for the suggestion. Actually, the convex or concave trend is dependent on the time-dependent solution of the extinction curve, but not the initial community configuration. That is, regardless of how species abundance distribution is for the initial community ($t = 0$), the curve shape (convex versus concave) is totally determined by the time-dependent solution of the net biodiversity loss (i.e., species richness at time t versus that at time zero).

It can be easily proved that the main equation 4 presented in Sgardeli et al. (2017)’s paper is a convex function:

$$S(t) = S_{eq} + \frac{2S_{eq}}{\frac{S_0 + S_{eq}}{S_0 - S_{eq}} e^{rt} - 1}.$$

To specifically demonstrate the above claim, one can show that the first derivative of the above equation with respect to t is

$$S'(t) = -\frac{2S_{eq}r \frac{S_0 + S_{eq}}{S_0 - S_{eq}}}{\left(\frac{S_0 + S_{eq}}{S_0 - S_{eq}} e^{rt} - 1\right)^2}.$$

One can see $S'(t)$ is always negative; meanwhile, $S'(t)$ is increasing with t . To this end, it can be concluded that Sgardeli et al. (2017)'s main equation is convex.

By contrast, the time-dependent solution in our paper actually, regardless of at the metacommunity level (Eq. 9 in the main text) or at the local area level, is a convex function when immigration is absent (i.e., $u = 0$). To demonstrate this, we use Eq. 9 in the main text as an example,

$$E(t|A) = \begin{cases} \sum_{j=1}^{\infty} S_j(0|A) \left(\frac{t}{1+t}\right)^j, & u = 0; \\ \sum_{j=1}^{\infty} S_j(0|A) \left(\frac{1 - e^{-ut}}{1 - (1-u)e^{-ut}}\right)^j, & u > 0. \end{cases}$$

When $u = 0$, the first and second derivatives of $E(t|A)$ are

$$\begin{aligned} E'(t|A) &= \sum_{j=1}^{\infty} S_j(0|A) \left(\frac{t}{1+t}\right)^{j-1} j \left(\frac{1}{1+t} - \frac{t}{(1+t)^2}\right) = \sum_{j=1}^{\infty} S_j(0|A) \left(\frac{t}{1+t}\right)^{j-1} j \frac{1}{(1+t)^2} \\ &= \sum_{j=1}^{\infty} S_j(0|A) j \frac{t^{j-1}}{(1+t)^{j+1}} \end{aligned}$$

and

$$E''(t|A) = \frac{d}{dt} \sum_{j=1}^{\infty} S_j(0|A) j \frac{t^{j-1}}{(1+t)^{j+1}} = \sum_{j=1}^{\infty} S_j(0|A) j \frac{(j-1)t^{j-2}}{(1+t)^{j+2}},$$

respectively. Since $E''(t|A)$ is always positive decreases, we prove that $E(t|A)$ is a convex function of t .

Note that the case for $u > 0$ follows a similar derivation as for $u = 0$; we found the curvilinear pattern of $E(t|A)$ is dependent on time t . Specifically, the second derivation of $E(t|A)$ is shown to have an analytical form as follows:

$$E''(t|A) = \sum_{j=1}^{\infty} S_j(0|A) j u^2 (u(1-u)e^{-2ut} + ju^2 e^{-ut} - u) \frac{e^{-ut}(1 - e^{-ut})^{j-2}}{[1 - (1-u)e^{-ut}]^{j+2}},$$

from which, it reveals that the curvilinear pattern of $E(t|A)$ is indeterministic.

As a remark, it is also possible to construct a concave trend pattern for the model of our paper when time is log-transformed. To demonstrate this, we simulate a curve with the following configurations: The local intact area of interest has a size of 1, while the total region has an area with a size of 100, initial species richness=1000, Parameter ω is set to 0.1, and the emigration parameter, u , is set to 0.001.

One can see that, some convex trend can be observed at small time interval from subplot B of the following figure (log time values smaller than 4) when X-axis (time) is log-transformed. As a comparison, the original curve shape without log-transformation is fully concave (Figure below).

Finally, because our intention is to compare the general species loss model, therefore, in the revised text, we compared our Eq. 8 in the revised paper to the following equation:

$$E_S(t) = S_0 - S(t) = S_0 - S_{eq} - \frac{2S_{eq}}{\frac{S_0 + S_{eq}}{S_0 - S_{eq}} e^{rt} - 1}$$

Apparently, because $S(t)$ is convex, $E_S(t)$ becomes concave. But as just mentioned above, when log-transforming the time scale, both concave and convex trends can be observed.

References

- Allen, L. 2010. An introduction to stochastic processes with applications to biology. - Chapman and Hall/CRC.
- Brown, J. and Kodric-Brown, A. 1977. Turnover rates in insular biogeography: effect of immigration on extinction. - Ecology 58: 445–449.
- Chen, Y. and Shen, T. 2017. A general framework for predicting delayed responses of ecological communities to habitat loss. - Scientific Reports: 998.
- Dornier, A. and Cheptou, P. 2012. Determinants of extinction in fragmented plant populations: *Crepis sancta* (Asteraceae) in urban environments. - Oecologia 169: 703–712.
- Eriksson, A. et al. 2014. The emergence of the rescue effect from explicit within- and between-patch dynamics in a metapopulation. - PRSB 281: 20133127.
- Gotelli, N. 1991. Metapopulation models: the rescue effect, the propagule rain, and the

- core-satellite hypothesis. - *American Naturalist* 138: 768–776.
- Halley, J. M. et al. 2016. Dynamics of extinction debt across five taxonomic groups. - *Nature Communications* 7: 12283.
- He, F. 2005. Deriving a neutral model of species abundance from fundamental mechanisms of population dynamics. - *Functional Ecology* 19: 187–193.
- Hufbauer, R. et al. 2015. Three types of rescue can avert extinction in a changing environment. - *PNAS* 112: 10557–10562.
- Kitzes, J. and Harte, J. 2015. Predicting extinction debt from community patterns. - *Ecology* 96: 2127–2136.
- Lawson, C. R. et al. 2012. Local and landscape management of an expanding range margin under climate change. - *Journal of Applied Ecology* 49: 552–561.
- Mills, L. and Allendorf, F. 1996. The one-migrant-per-generation rule in conservation and management. - *Conservation Biology* 10: 1509–1518.
- Rosindell, J. et al. 2010. Protracted speciation revitalizes the neutral theory of biodiversity. - *Ecology Letters* 13: 716–727.
- Sgardeli, V. et al. 2017. A forecast for extinction debt in the presence of speciation. - *Journal of Theoretical Biology* 415: 48–52.
- Spielman, D. and Frankham, R. 1992. Modeling problems in conservation genetics using captive *Drosophila* populations: improvement of reproductive fitness due to immigration of one individual into small partially inbred populations. - *Zoo Biology* 11: 343–351.
- Stacey, P. and Taper, M. 1992. Environmental variation and the persistence of small populations. - *Ecological Applications* 2: 18–29.
- Volkov, I. et al. 2005. Density dependence explains tree species abundance and diversity in tropical forests. - *Nature* 438: 658–661.

Appendix C

Response Letter

11-Nov-2019

Dear Professor Shen,

On behalf of the Editors, I am pleased to inform you that your Manuscript RSOS-191039.R1 entitled "Extinction debt in local habitats: quantifying the roles of random drift, immigration, and emigration" has been accepted for publication in Royal Society Open Science subject to minor revision in accordance with the referee suggestions. Please find the referees' comments at the end of this email.

The reviewers and Subject Editor have recommended publication, but also suggest some minor revisions to your manuscript. Therefore, I invite you to respond to the comments and revise your manuscript.

- Ethics statement

- Data accessibility

<http://datadryad.org/submit?journalID=RSOS&manu=RSOS-191039.R1>

- Competing interests

- Authors' contributions

All submissions, other than those with a single author, must include an Authors' Contributions section which individually lists the specific contribution of each author. The list of Authors should meet all of the following criteria; 1) substantial contributions to conception and design, or

acquisition of data, or analysis and interpretation of data; 2) drafting the article or revising it critically for important intellectual content; and 3) final approval of the version to be published.

- Acknowledgements

- Funding statement

Because the schedule for publication is very tight, it is a condition of publication that you submit the revised version of your manuscript before 20-Nov-2019. Please note that the revision deadline will expire at 00.00am on this date. If you do not think you will be able to meet this date please let me know immediately.

- 1) A text file of the manuscript (tex, txt, rtf, docx or doc), references, tables (including captions) and figure captions. Do not upload a PDF as your "Main Document".

- 2) A separate electronic file of each figure (EPS or print-quality PDF preferred (either format should be produced directly from original creation package), or original software format)
- 3) Included a 100 word media summary of your paper when requested at submission. Please ensure you have entered correct contact details (email, institution and telephone) in your user account
- 4) Included the raw data to support the claims made in your paper. You can either include your data as electronic supplementary material or upload to a repository and include the relevant doi within your manuscript
- 5) All supplementary materials accompanying an accepted article will be treated as in their final form. Note that the Royal Society will neither edit nor typeset supplementary material and it will be hosted as provided. Please ensure that the supplementary material includes the paper details where possible (authors, article title, journal name).

Kind regards,

on behalf of Dr Punidan Jeyasingh (Associate Editor) and Kevin Padian (Subject Editor)
openscience@royalsociety.org

Associate Editor Comments to Author (Dr Punidan Jeyasingh):

I thank the authors for a thorough revision of the original manuscript based upon highly constructive expert reviewer comments. This version was reassessed by these experts, both of whom were largely satisfied with the revisions. Nevertheless, they have raised a few issues that I felt were fair and constructive. I invite the authors to make these final adjustments, and look forward to seeing this paper in print.

Dear Editors,

Thanks for your encouraging comments and again providing a chance for us to revise the

paper according to the constructive suggestions of the two reviewers. Consistently, we always value the hard-won opportunity and try our best to thoroughly revise the paper along with a one-to-one response letter below for your perusal. As usual, our responses are highlighted in *blue* color while the changes made in the paper and appendix are underscored in *red* color.

We hope our substantial efforts for the revision after incorporating your constructive comments allow the paper to become more valuable, readable, and attractive to the readership of the journal after publication.

Yours Sincerely,
The authors

Reviewer comments to Author:

Reviewer: 1

Comments to the Author(s)

I think the authors have done a great effort to improve the manuscript and solve the critical issues, as raised in my previous reports. Before the acceptance of the paper, the authors should address some minor points:

Response: Thanks for the encouraging comments.

1) The y-axis label must be changed in the figures. In fact, the correct is "number of species" (or biodiversity level) instead of total species loss. The species loss corresponds to the gap between curves before and after disturbance.

Response: thanks for the suggestion, in the revision, we updated the y-axis label in the main text and supporting information as "number of species" as suggested. Additionally, to make it more specific, we also gave an interpretation in the figure captions as "The y-axis denotes the difference of the number of species at the initial time and that at a specific time point afterwards".

2) The authors should compare their outcomes with those already shown in the literature. As examples of studies on this issue please see Claudino et al., Extinction debt and the role of static and dynamical fragmentation on biodiversity. *Ecological Complexity* (Print), v. 21, p. 150-155, 2015; Gomes et al. Effect of Landscape Structure on Species Diversity. *Plos One*, v. 8, p. e66495, 2013; Thompson et al., Characterising extinction debt following habitat fragmentation using neutral theory, *Ecol. Letters* (2019); Coelho Neto et al., Neutral communities in fragmented landscapes. *Oikos* (Kobenhavn), v. 121, p. 1737-1748, 2012.

Response: thanks for providing us with a great deal of helpful literature related to the topic we studied, all of which are important and valuable for showing the close relationship between distributional aggregation and species diversity and extinction. In the revision, we mentioned, discussed and compared our results and models with respect to all these preceding references as suggested (and some other references, e.g., [1]). To be specific, for further research, we discussed the necessity of incorporating spatial aggregation on the modeling of time-related species diversity and extinction patterns and the degree of habitat fragmentation in the last paragraph of the discussion section of the revised main text.

Reviewer: 2

Comments to the Author(s)

[REVISED PAPER] Journal, Manuscript ID: Royal Society Open Science - RSOS-191039

By Y Chen, S-C Chang, Y Wu, Y-F C, T-J Shen

The authors have addressed all of my concerns seriously and carefully. Only one thing remains, though. Regarding my 9th point: “The ‘convex trend’ in Figs 1-3 prior to habitat loss, is probably an increase in diversity due to starting diversity at a low value (smaller than equilibrium). The authors should allow the system time to reach equilibrium prior to the habitat destruction. An extra figure might explore the convex or concave trends for different starting values of biodiversity.’

To this the authors reply “Actually, the convex or concave trend is dependent on the time-dependent solution of the extinction curve, but not the initial community configuration. That is, regardless of how species abundance distribution is for the initial community ($t = 0$), the curve shape (convex versus concave) is totally determined by the time-dependent solution of the net biodiversity loss (i.e., species richness at time t versus that at time zero) ... One can see is always $S'(t)$ negative; meanwhile, $S'(t)$ is increasing with t . To this end, it can be concluded that Sgardeli et al. (2017)’s main equation is convex.”

But wait. The time-dependent solution of the extinction curve contains S_0 , the initial species richness. Note that $S'(t)$ will have a different sign according to the factor ($S_0 - Seq$). Thus $S'(t)$ can be positive or negative according to the initial diversity S_0 . This makes sense. If we start above equilibrium, diversity must diminish and if we start below it must increase.

Response: thanks for your insightful comment that we totally agreed on. We unintentionally took no account of the sign of the factor $S_0 - Seq$ in the previous revision; thanks for clarifying this issue to us. In the revised main text and appendix, we did clearly mention the sign issue caused by the term $S_0 - Seq$. Moreover, we emphasized that, in most cases, when studying extinction debt, the $S_0 - Seq$ is positive because the initial species richness (S_0) is usually larger than species richness at the equilibrium (Seq), and consequently, the convex curve can be observed.

But my point is that Fig1 as it stands is confusing because we need more ecological context.

Thus,... insert a second panel into Fig 1 (not an Appendix) to show $S(t)$ as a function of time, and the problem is solved. This will help the readers greatly.

Response: thanks for the suggestion. In the revision, we inserted subplot B into Fig. 1 of the revised text for showing the dynamic of $S(t)$ over time for the main models studied in the paper.

References

1. Chisholm R, Lim F, Yeoh Y, Seah W, Condit R, Rosindell J. 2018 Species-area relationships and biodiversity loss in fragmented landscapes. *Ecology Letters* 21, 804–813.